# Knowledge, attitude, and practice of seasonal influenza and influenza vaccine immunization among people visiting primary healthcare centers in Riyadh, Saudi Arabia

**Norah Alhatim**[1☯], **Ahmad M. Al-Bashaireh**[2☯], **Ola Alqudah** [3]*

**1** King Fahad Medical City Academy for Postgraduate Studies in Family Medicine, Riyadh, Kingdom of Saudi Arabia, **2** Department of Primary Care Nursing, Faculty of Nursing, Al-Ahliyya Amman University, Amman, Jordan, **3** Department of Community Health, King Fahad Medical City, Riyadh, Kingdom of Saudi Arabia

☯ These authors contributed equally to this work.
* drola1980m@gmail.com

**Data Availability Statement:** Concerning data, there are no legal or ethical restrictions on sharing our data publicly and uploaded in figshare

## Abstract

Influenza infection continues to be a hazard to the Saudi population, resulting in high death rates and illness prevalence; it also places a substantial financial burden on the government. The government takes several strategies and approaches through the Ministry of Health has shown great success in curbing the disease. Vaccination is considered the most appropriate control measure; unfortunately, most Saudi residents, particularly in the city of Riyadh, do not consider vaccination a safe health practice. As a result, many have not participated in the influenza vaccine immunisation programme. Therefore, this study aimed to assess the knowledge, attitudes, and practices (KAPs) of seasonal influenza and influenza vaccine immunisation among clients visiting primary healthcare centers in Riyadh, Saudi Arabia. Furthermore, the study investigated the relationship between participants' demographics and their KAPs regarding influenza vaccination. A cross-sectional, descriptive, correlational study was conducted among 611 individuals who visited four of Riyadh's primary healthcare centers: Alsylimania, Alwady, Alyasmin, and Alsahafah. A self-reported questionnaire was used to assess the KAPs of participants, with questions regarding seasonal influenza and influenza vaccine immunisation. The scores of participants' knowledge showed that the majority had good knowledge regarding seasonal flu (64.5%) and the flu vaccine (73.3%). Furthermore, only 52% of participants had a positive attitude score towards the seasonal influenza vaccination. Despite that, significant knowledge gaps and mistaken beliefs regarding certain aspects of influenza were noted in participants, resulting in negative attitudes and perceptions as well as a reduced likelihood of being vaccinated. In this study, 43.7% of participants (267 out of 611) had ever received a flu vaccine. Participants with a history of previous vaccination had a significantly higher level of knowledge and more positive attitudes, which resulted in increased vaccination coverage. Therefore, educational strategies to improve knowledge regarding influenza in Riyadh are recommended.

repository https://figshare.com/s/c49083030fc8bf9f6ed7.

**Funding:** We did not received any funding for this research.

**Competing interests:** The authors have declared that no competing interests exist.

## Introduction

According to the World Health Organization, Influenza A and B are responsible for seasonal flu epidemics across many parts of the world each year; influenza affects people of all age groups, with a common trend of 5–10% and 20–30% in adults and children, respectively [1]. The Spanish influenza of 1918–1919 was among the first and most tragic historical pandemics, reportedly causing infection in more than one-third of the world's population [2]. The Centers for Disease Control and Prevention has reported other influenza-related cases in recent years, with the most recent pandemic in 2009 caused by the Influenza A virus (H1N1, pdm09) [3]. Influenza infections across the world affect an average of five million people, with close to 500,000 deaths globally each year, most of which are children (<12 years of age) or the elderly (>65 years of age) [1].

Due to hospital admissions and productivity loss, influenza is responsible for enormous economic costs [4]. It is an acute disease that targets the upper respiratory tract, resulting in inflammation as the body works to rapidly deliver immune cells to the infection site [5]. The immune system reacts, releasing cytokines and chemokines (interferon), which cause symptoms such as high fever, coryza, and body aches [6]. Influenza A, the most common type in humans, is a genetically labile virus; compared to other microbes, Influenza A's mutation rate is over 300 times faster than influenza B [7]. This is due to changes in its primary functional and antigenic proteins, which occur by two mechanisms: antigenic drift and antigenic shift [7].

Therefore, influenza vaccines are recommended by most healthcare providers and organisations so that people can be protected against this infection—not just medically but also economically and socially. Vaccines work by ensuring the body develops antibodies about two weeks after administration, which protect against infection [3]. Duque et al. (2014) explain that vaccination is the most effective method of preventing influenza; its development has evolved over the years due to mutations in virus structures and new emerging strains [8]. The goal of the influenza vaccine is to protect against disease; recent research advances have focussed on creating a universal vaccine that offers protection against all influenza virus strains, addressing the issue of antigenic drift and shifts [9].

Many regions worldwide experience influenza; the Middle East is no exception. Influenza remains an extreme threat in Saudi Arabia, which was one of the most affected countries during the 2009 epidemic. Almost 100 cases were reported in 2010, with 124 deaths [10]. To date, there have been several other cases of influenza reported in Saudi Arabia, particularly in major cities and provinces across the country. This situation is exacerbated due to the massive yearly congregations of Muslims into the holy cities of Makkah and Madinah for Omera and Haj, when influenza strains may potentially be transported to the country [11]. The city of Riyadh serves as the capital city of Saudi Arabia; it's set in an urban population that has contributed to it being among one of the most affected regions of the country. Due to widespread cases and threats, the government under the Ministry of Health (MOH) has initiated several health precautions and strategies to prevent influenza pandemics, both at present and in the future. Vaccine immunisation is among these initiatives, with plans in place to offer immunisation at several health facilities in the city and across parts of the country. In addition, the Saudi MOH offers a free-of-charge influenza vaccine every year to anyone older than six months.

Furthermore, the Saudi MOH mandates an annual intake of influenza vaccination for all healthcare providers [12, 13]. However, the current vaccination rates are low, as many people base their decisions on being vaccinated based on religious and cultural beliefs. According to Sagor and AIAteeq (2018), a person's decision to receive the influenza vaccine depends on several factors, which can be categorised into knowledge, attitudes, and practices (KAPs)

concerning both influenza and the influenza vaccine [14]. The following paragraphs explore the KAPs concerning flu and its vaccines in the general population, healthcare workers (HCWs), and parents in Saudi Arabia.

A systematic review of 48 articles, with the majority from Saudi Arabia, revealed a knowledge gap regarding influenza and its vaccine among the public and healthcare workers. Also, this review revealed that lack of knowledge is the chief barrier to influenza vaccination [15]. A community-based cross-sectional study that enrolled 778 Saudi citizens showed gender and age group differences in the knowledge of influenza and influenza vaccination [16]. Female and lower age groups were found to have lower levels of knowledge of influenza and influenza vaccines [16]. Another cross-sectional study involved 790 Saudi citizens from the general population and showed that the participants who believed the influenza vaccine to be safe, efficacious, given at a specific time of the year and were aware of the need to be vaccinated were more likely to have received the vaccine [17]. A study by Alabbad et al. (2018), which enrolled three groups (adult patients, parents, and HCWs) of 300 Saudis, showed that the most common reasons for vaccination were awareness campaigns and being medical staff (36%). The most common reasons given by those who refused the vaccine were due to their beliefs that the vaccine had no benefit (21%), they were healthy, and so a vaccine was not needed (17%), and that the vaccine caused serious adverse effects (13%) [18]. A cross-sectional study of 496 Saudi participants aged 65 years and older showed that doctors and HCWs were the main sources of information about the influenza vaccination. In this study, only 40% of participants considered the influenza vaccine to be safe and effective [12].

Many other studies enrolled general population from Saudi Arabia, such as Sagor and AlAteeq (2018) and Alqahtani et al. (2017). Concerning the KAPs in Saudi Arabia, Sagor and AlAteeq (2018) conducted a study in the city of Riyadh and revealed most people in the country were not vaccinated, although those who were vaccinated were more likely to be men [14]. Additionally, historical knowledge contributed substantially to responses and decisions related to influenza vaccination. Many individuals with prior knowledge of the influenza vaccination were more likely to receive the vaccination than those with little or no knowledge; furthermore, the knowledge that hospitals were offering free vaccinations resulted in more individuals receiving their immunisations [14]. The study also revealed that people tended to respond to given health recommendations if they were informed and acquainted with relevant knowledge in advance [14].

According to Alqahtani et al. (2017), knowledge regarding vaccine immunisation is substantially associated with how people make decisions. The study revealed that vaccinated participants showed higher knowledge levels than non-vaccinated participants [11]. Also, individuals who had some prior form of interaction with the vaccine demonstrated better awareness, with a more in-depth understanding of how it affected their immune system; this was especially true when compared to those who had never been vaccinated, which was primarily due to negative perceptions that the vaccine weakens the immune system [11]. Of the participants who participated in the study, only 9% were aware of the fact that pregnant women could be vaccinated against influenza [11]. Another factor concerning the participants' level of knowledge was that most of them did not know that the government offered free vaccination in various health centers across the country [11].

A Saudi study among HCWs reported that at least 67% of them were vaccinated [19]. Most HCWs (84%) had a strong belief that the influenza vaccine helped to prevent influenza, with 75% believing they were more susceptible to these infections than other vulnerable groups; however, many respondents had concerns regarding the vaccine's safety, which was seen as the main barrier to vaccination [19]. Almost 42% of HCWs expressed the misconception that the vaccine contributed to influenza infection, with most displaying incorrect perceptions

regarding the symptoms and signs of the condition [19]. Another cross-sectional study involving 312 primary HCWs in Saudi Arabia showed that 45.5% of participants were vaccinated. Around one-third and a quarter of participants were found to show a lack of knowledge about influenza and the influenza vaccine, respectively [20]. Participants' awareness of their risk of infection and their need for protection was the main motivator (77.5%), while the fear of adverse effects was the main barrier to their receiving a vaccination (40%) [20].

Concerning the KAPs of parents towards influenza vaccination, a study by Alolayan et al. (2019) showed that the majority of Saudi parents (94.7%) had positive attitudes towards the influenza vaccine; however, the majority (61.7%) showed poor knowledge about the vaccine itself [21].

Influenza infection continues to be a hazard to the Saudi population, resulting in high death rates and illness prevalence; it also places a substantial financial burden on the government. Several strategies and approaches are taken by the government through the MOH have shown great success in curbing the pandemic. Vaccination is considered the most appropriate measure to control the disease; unfortunately, the majority of Saudi residents, particularly in the city of Riyadh, do not consider vaccination to be a safe health practice [19]. As a result, many have not participated in the influenza vaccine immunisation programme. Understanding the KAPs of participants will help in creating ways to improve influenza vaccination rates in Riyadh, minimising the impact and severity of seasonal influenza. Hence, this study aimed to assess the KAPs regarding seasonal influenza and influenza vaccine immunisation among clients visiting healthcare centers in Riyadh, Saudi Arabia. Furthermore, the study would investigate the relationship between patients' demographics and their KAPs regarding influenza vaccination.

## Materials and methods

This study employed a cross-sectional, descriptive, correlational design. The study was conducted in four primary healthcare centers in Riyadh, Saudi Arabia. One of the significant considerations of the study was to assess people who attend these centers either for treatment, bringing family members for treatment, or visiting. Additionally, the study engaged pregnant women and parents with young children. A sample of 611 individuals was recruited between January 15 and July 5, 2020. The inclusion criteria were participants willing to be part of the study, fluent in the Arabic language, and 18 years and older.

Active recruitment was implemented in this study. Researchers contacted potential participants from the four primary healthcare centers in Riyadh city (Alsylimania, Alwady, Alyasmin, and Alsahafah). They explained the purpose, confidentiality, anonymity, and voluntariness of participating in this study. Potential participants were anyone who visited these four primary healthcare centers 18 years old and above, can read and write the Arabic language, including patients or their companions (e.g., parents of young children and family caregivers of elderly patients). If participants gave their verbal consent, they were asked to fill a self-reported questionnaire that includes demographics, knowledge, and attitude concerning seasonal influenza and influenza vaccination. Participants took a time of 10 to 15 minutes to complete the self-reported questionnaire.

A validated questionnaire had been used in a previous study [22], but on a different and new population. The questionnaire was available online and used under the terms and conditions of Attribution-NonCommercial-NoDerivatives 4.0 International (CC BY-NC-ND 4.0). The original questionnaire was in the English language and was used to evaluate the knowledge, attitudes, and practices regarding seasonal influenza and influenza vaccination among patients with diabetes mellitus [22]. Three items from the original questionnaire were

removed. These items were: (1) How long had you known that you are diabetic? (2) What do you know about seasonal flu: (a) flu symptoms are worse among people with diabetes and (b) cause serious complications among diabetics; and (3) flu can cause severe complications among those who have diabetes.

After the removal of non-pertinent items from the original questionnaire, the questionnaire was translated into Arabic and then back-translated into English, while the contents were validated by a group of experts, including physicians, nurses and epidemiologists. The Arabic versions of this questionnaire were piloted on 30 subjects, and it was checked for clarity and understandability by the participants. In general, the questionnaire was clear and showed an acceptable level of internal reliability (Cronbach's-alpha > 0.70) for the three domains of knowledge, attitudes, and practices.

In our study, the questionnaire covered four major domains: demographics (5 items), knowledge (11 major items), attitudes (7 major items), and practices (4 major items). Knowledge, attitudes, and practices (KAP) domains have major questions followed by a set of minor questions. Participants' answers in KAP domains were multiple-choice questions, and the participants ticked for the appropriate answers. The major questions of these domains were listed in Box 1.

Interpretation of the questionnaire's findings: The same cut-offs as those used by Olatunbosun et al. (2017) were used when interpreting the scores of each domain of the questionnaire [22]. Regarding Knowledge (of seasonal influenza and influenza vaccine immunisation), based on questions that were answered correctly, a score of 65% and above was graded as 'good', while a score below 65% was considered as 'poor'. Attitudes were categorised as 'positive' at four or more positive responses or as 'negative' for four or more negative responses to the Attitudes questions. For the measure of Practices, the study considered any two positive responses as 'good', which represented 66.3%, while two negative responses were categorised as 'poor' based on the answers to the questions in the Practices section.

Researchers obtained approval from the IRB at the King Fahad Medical City (IRB00010471) and the administrators of Alsylimania, Alwady, Alyasmin and Alsahafah healthcare centers in Riyadh. Verbal informed consent was obtained from all participants. Before the commencement of the questionnaire, all participants were informed about confidentiality, anonymity and voluntary participation, and they were required to give a verbal acknowledgement regarding their consent and understanding of the legal terms applicable to the study. All information obtained from participants was kept private and confidential.

Sample size was calculated using standard online tools through the following formula: $N = (Z\alpha)^2 \times ([p(1 -p)]/d^2)$; where: n = estimated sample size, $Z\alpha$ at 5% level of significance = 1.96, d = level of precision, estimated to be 0.05, p = high awareness levels in two previous studies (30%); hence, the primary sample size = $[(1.96)^2 \times (0.3 \times 0.7)]/(0.05 \times 0.05)$ = 329 subjects. Actual sample size = (Primary sample size × design effect (estimated to be 1.5) = 493 subjects. The expected response rate was estimated to be 80%. Therefore, the planned sample size = 493 × 100 / 80 = 616 subjects.

Data entry and other statistical analyses were performed with SPSS version 22, aiming at a significant difference of ≤0.05. Frequencies and percentages were used to describe categorical variables and mean, and standard deviation were used to describe continuous quantitative variables. Chi-square and Fishers-exact tests were employed to compare the categorical outcomes. Descriptive statistics were used to describe the findings of the KAPs regarding seasonal influenza and influenza vaccination. Furthermore, a logistic regression model was used to explore factors associated with previous vaccination (vaccinated at some point versus never vaccinated).

Box 1. Major items of the four domains.

I. **Socio-demographics**

  1. How old are you?

  2. What is your gender?

  3. What is your marital status?

  4. What is your current occupation?

  5. What is your level of education?

II. **Knowledge**

  1. What do you know about seasonal influenza?

  2. What are the symptoms of flu that you know?

  3. Have you ever heard of that a vaccine could prevent flu?

  4. Does the vaccine prevent the flu?

  5. How is the vaccine given?

  6. Does the vaccine have side effects?

  7. What are the side effects of the flu vaccine

  8. How long vaccine can protect you?

  9. Does the influenza vaccine can prevent complication associated with seasonal flu?

  10. When is the appropriate time to take the vaccine?

  11. Is it true that you can never have flu so long as you are vaccinated during the seasonal flu?

III. **Attitudes**

  1. Influenza vaccine is important and should be taken yearly.

  2. Influenza vaccination prevents complications associated with seasonal flu.

  3. Influenza vaccine has a serious side effects, therefore should not be taken.

  4. All people should take the influenza vaccine.

  5. Flu is a mild illness and therefore vaccination is not necessary.

  6. Flu is a mild illness and therefore vaccination is not necessary.

  7. I don't need the flu vaccine because I have life immunity against the flu.

IV. **Practices**

  1. Have you ever received the influenza vaccine before?

2. How regularly do you take the influenza vaccine?

3. What influenced you to take the vaccine?

4. What are the reasons for not taking the influenza vaccination (answered if the first question is no)?

## Results

Our study enrolled 611 participants with an average age of 36.2 ± 12.1 years. The majority were male (63.5%), married (55.6%), had a job (70.9%) and a college/university academic degree (76.3%; Table 1). Participants' knowledge concerning seasonal influenza revealed that the majority believed that flu was caused by a virus (89.4%), can spread from one to another (96.1%) and occurs at a certain period of the year (76.1%). Interestingly, only 57% of them believed that flu could be prevented and 32.6% believed that seasonal flu was similar to the common cold (Table 2). The most frequently reported symptoms were running nose (95.7%), fever (89.5%), sore throat (87.7%), sneezing (82.5%), cough (82.3%) and headache (70%; Table 2).

In our sample, only 499 out of 611 (81.7%) reported that they heard of a vaccine that prevented flu. Further questions were asked concerning the flu vaccine for those who had heard about the vaccine (Table 3). Of those, 71.5% believed that the vaccine was safe and only 49.1% reported that the vaccine could prevent flu. Most participants indicated that this vaccine was given via injection and only a few percent said that it could be given via nasal spray and orally: 6.2% and 10%, respectively. The majority of participants (98.8%) said that the vaccine had side effects and that the most frequent side effects were soreness/swelling at the injection site (83%), fever (79.8%) and muscle ache (60%). The majority of participants believed that the

**Table 1. Socio-demographic characteristics (N = 611).**

| Characteristics | Mean ± SD or Frequency (%) |
|---|---|
| **Age** | 36.2 ± 12.1 |
| **Sex** | |
| Male | 388 (63.5) |
| Female | 223 (36.5) |
| **Marital Status** | |
| Single | 177 (29.0) |
| Married | 340 (55.6) |
| Separated | 11 (1.8) |
| Divorced | 67 (11.0) |
| Widow | 16 (2.6) |
| **Occupation** | |
| Working | 433 (70.9) |
| Not working | 163 (26.7) |
| Retired | 15 (2.5) |
| **Level of Education** | |
| Basic | 7 (1.1) |
| Elementary/Secondary | 138 (22.6) |
| College/University | 466 (76.3) |

**Table 2. Participants' knowledge regarding seasonal influenza (N = 611).**

|  | Frequency (%) |
|---|---|
| **Seasonal influenza** |  |
| Flu is caused by a virus | 546 (89.4) |
| Flu can spread from one person to another | 587 (96.1) |
| Flu can be prevented | 354 (57.9) |
| Flu is the same as a common cold | 199 (32.6) |
| Flu occurs at a certain period of the year | 465 (76.1) |
| **Symptoms** |  |
| Running nose | 585 (95.7) |
| Sneezing | 504 (82.5) |
| Headache | 428 (70.0) |
| Sore throat | 536 (87.7) |
| Cough | 503 (82.3) |
| Vomiting | 163 (26.7) |
| Fatigue | 483 (79.1) |
| Muscle ache | 397 (65.0) |
| Fever | 547 (89.5) |
| Diarrhea | 145 (23.7) |
| Abdominal pain | 130 (21.3) |

vaccine gave protection for one year/season; however, a small percentage of participants believed that the vaccine could protect for two years (8.6%) or for three years/seasons (9.2%). Only 59.3% of participants believed that the vaccine prevented complications associated with seasonal flu. A majority of participants reported that the vaccine should be taken before the beginning of the flu season (74.6%), while 22.4% and 3% of participants indicated that it should be taken during or immediately after the flu season, respectively, while 38.7% of participants believed that vaccination during the flu season would prevent infection (Table 3).

The scores of participants' knowledge showed that the majority had good knowledge about seasonal flu (64.5%) and the flu vaccine (73.3%), with an overall combined knowledge score regarding seasonal flu and the vaccine of 71.1% (Table 4). Table 4 provides further details about the means and medians of participants' knowledge scores.

Concerning the attitudes of participants towards the flu vaccination, most participants agreed about the importance of the vaccine, which should therefore be taken on a yearly basis (52.9%) and only 47% would recommend the flu vaccine for all people. Further, most of the participants agreed that the influenza vaccine prevented serious complications (52.7%) and the majority would take the vaccine if it were effective in preventing seasonal flu (75.6%). Most of the participants disagreed with the idea that flu is a mild illness, and that vaccination was not necessary (54.3%) and also disagreed that they did not need a flu vaccine because they were immune against infection (40.4%). Finally, only 36.2% agreed that the vaccine had serious side effects and should not be taken (Table 5). Positive attitude scores were shown by 52% of participants, while 48% had a negative attitude towards the seasonal influenza vaccination.

A few percent of participants reported that they had been admitted to a hospital due to flu infection (11.5%). There were 43.7% of participants (267 out of 611) that had ever received a flu vaccine, either annually (44.2%), every two years (19.9%) or every three years (29.2%; Table 6). The most frequent factors that influenced previous vaccination decisions were doctor's advice (63.3%), the availability of free charge vaccine (52.4%) and a recommendation from other patients about vaccine effectiveness (27.7%). Furthermore, 30 participants (11.2%)

**Table 3. Participants' knowledge regarding flu vaccine (N = 499).**

| Characteristics | Frequency (%) |
|---|---|
| **Is the flu vaccine safe?** | 357 (71.5) |
| **Does the vaccine prevent the flu?** | 245 (49.1) |
| **How is the vaccine given, check all apply?** | |
| Injection | 494 (99.0) |
| Nose spray | 31 (6.2) |
| Mouth drops | 50 (10.0) |
| **Does the vaccine have side effects?** | 493 (98.8) |
| **If participants answer yes for side effects, what is it?** | |
| Soreness/swelling at the injection site | 414 (83.0) |
| Fever | 398 (79.8) |
| Muscle ache | 299 (60.0) |
| Headache | 208 (41.7) |
| Nausea | 176 (35.3) |
| Other symptoms | 24 (4.8) |
| **For how long can the vaccine protect?** | |
| 1 year/season | 410 (82.2) |
| 2 years/seasons | 43 (8.6) |
| 3 years/seasons | 46 (9.2) |
| **Does the influenza vaccine can prevent complication associated with seasonal flu?** | 296 (59.3) |
| **When is the appropriate time to take the influenza vaccine?** | |
| Before flu season starts | 372 (74.6) |
| During the flu season | 112 (22.4) |
| Immediately after flu season | 15 (3.0) |
| **You can never have flu as long as you vaccinated during the seasonal flu?** | 193 (38.7) |

**Table 4. Participants' knowledge score.**

| Score | N | Mean ± SD | Median (Range) | Frequency (%) of good level | Frequency (%) of poor level |
|---|---|---|---|---|---|
| Seasonal flu knowledge score | 611 | 11.17 ± 2.29 | 11.00 (3–16) | 394 (64.5) | 217 (35.5) |
| Flu vaccine knowledge score | 499 | 13.92 ± 2.11 | 14.00 (7–20) | 368 (73.7) | 131 (26.3) |
| Total knowledge score (seasonal influenza + flu vaccine) | 499 | 25.29 ± 3.28 | 25.00 (15–36) | 355 (71.1) | 144 (28.9) |

**Table 5. Participants' attitudes regarding influenza vaccination (N = 611).**

| | Agree Frequency (%) | Disagree Frequency (%) | Don't Know Frequency (%) |
|---|---|---|---|
| Influenza vaccination is important and should be taken yearly | 323 (52.9) | 139 (22.7) | 149 (24.4) |
| Influenza vaccine prevent serious complication associated with seasonal influenza | 322 (52.7) | 170 (27.8) | 119 (19.5) |
| Influenza vaccine has a serious side effect, therefore should not be taken | 221 (36.2) | 252 (41.2) | 138 (22.6) |
| All people should receive influenza vaccine | 287 (47.0) | 153 (25.0) | 171 (28.0) |
| Flu is a mild illness and therefore vaccination is not necessary | 176 (28.8) | 332 (54.3) | 103 (16.9) |
| I don't need the flu vaccine because I have life immunity against flu | 205 (33.6) | 247 (40.4) | 159 (26.0) |
| If there is an effective vaccine to prevent seasonal flu, I will take it | 462 (75.6) | 69 (11.3) | 80 (13.1) |

**Table 6. The previous history concerning hospitalization due to flu infection, receiving a flu vaccination, and frequency of vaccination (total number = 611, ever vaccinated = 267).**

| | Frequency (%) |
|---|---|
| Admission to the hospital due to flu infection | 70 (11.5) |
| Have you received the influenza vaccine before? | 267 (43.7) |
| For those who vaccinated (n = 267), how regularly do they take the vaccine? | |
| Yearly | 118 (44.2) |
| Every 2 years | 53 (19.9) |
| Every 3 years | 78 (29.2) |
| Other | 18 (6.7) |

**Table 7. Reasons were given by participants for not receiving previous influenza vaccine (N = 344).**

| Reasons | Frequency (%) |
|---|---|
| I have alternative protection | 201 (58.4) |
| It has a serious side effect | 190 (55.2) |
| The vaccine is not effective | 163 (47.4) |
| It is not necessary because flu is just a minor illness | 155 (45.1) |
| People who got the vaccine before is immune | 94 (27.3) |
| It is expensive | 91 (26.5) |
| Fear of needles and injection | 90 (26.2) |
| I reacted to at the first time I attempted it | 25 (7.3) |

said that other reasons had motivate or forced them to get the flu vaccine, such as educational campaigns conducted in shopping malls, schools, and universities (30%), employment requirements (23.3%), preparation by Muslim pilgrims (Alhaj) (16.7%), media and television (6.7%) and miscellaneous reasons (23.3%). Other reasons for vaccination are not shown in the table. Table 7 shows the reasons that prevented participants from taking the flu vaccine. The main reasons being indication by participants that they had alternative protection from flu (58.4%), the vaccine had serious side effects (55.2%), the vaccine was not effective (47.4%) and the vaccine was not necessary since flu is a minor illness (45.1%).

Table 8 compares the knowledge and attitudes between those who had received the vaccine at some point and those who had never been vaccinated regarding seasonal flu and flu

**Table 8. Knowledge and attitudes pertaining to influenza vaccination history (N = 611).**

| Characteristics | Vaccinated Frequency (%) n = 267 | Not vaccinated Frequency (%) n = 344 | p-value |
|---|---|---|---|
| Believes influenza vaccine is safe | 231 (86.5) | 126 (36.6) | < 0.001 |
| Believes influenza vaccine work to prevent flu | 159 (59.6) | 86 (25) | < 0.001 |
| Believes influenza vaccine has a side effect | 177 (66.3) | 188 (54.7) | 0.002 |
| Believes influenza vaccine can protect for only one flu season | 224 (83.9) | 186 (54.1) | 0.064 |
| Believes influenza vaccine can prevent serious complication among people | 182 (68.2) | 140 (40.7) | < 0.001 |
| Believes influenza vaccination is important and should be taken yearly | 199 (74.5) | 124 (36.1) | < 0.001 |
| Disagrees that influenza vaccine has a serious side effect and therefore should not be taken | 160 (59.3) | 92 (26.7) | < 0.001 |
| Would take influenza vaccine to prevent if effective | 222 (83.1) | 240 (69.8) | < 0.001 |
| Would recommend the influenza vaccine to all people | 190 (71.2) | 97 (28.2) | < 0.001 |

vaccination. In general, participants with a history of previous vaccination had a significantly higher level of knowledge and positive attitudes. The multivariate logistic regression showed age and attitude towards the flu vaccination were significantly and independently associated with the history of previous vaccination when controlling for other factors in the model. Given that other factors were controlled for, participants with higher age by 1 year were 1.6 more likely to receive the flu vaccine (OR = 1.016, 95%CI: 1.001–1.031, p = 0.035). Moreover, participants with a positive attitude were 5.579 times more likely to receive an influenza vaccine than those with negative attitudes (OR = 5.579, 95%CI: 3.906–7.969, p < 0.001); given that other factors were controlled for.

## Discussion

The primary aim of this study was to assess the KAPs of seasonal influenza and influenza vaccine immunisation among people visiting primary healthcare centers in Riyadh, Saudi Arabia. A literature review concerning influenza and influenza vaccination in Riyadh, as well as Saudi Arabia as a whole, was conducted. By undertaking questionnaires at four of the local healthcare facilities in Riyadh, 611 participants, who averaged 36.2 years of age, were enrolled in the current study. As the results indicate, there are negative perceptions and attitudes existing in Saudi Arabia, with many individuals suffering from a lack of knowledge regarding the influenza virus and vaccination. Most of the participants questioned were married men who had a university degree as well as gainful employment. The results of the questionnaires indicated they had good knowledge regarding seasonal influenza. Almost 90% of respondents understood that the flu was caused by a virus, with over 96% knowing that it spread easily among people in close contact; however, only a little more than half believed it could be prevented, with almost one-third mistakenly thinking that flu was similar to the common cold. There were 81.7% who knew that a vaccine was available to prevent flu; 71.5% thought it was safe, while just less than half thought that it was not effective. In a similar study (e.g., on the same population), Aljamili et al. (2020) revealed that 86.9% of participants were of the opinion that flu was a highly contagious disease that may require hospitalisation [16]. Moreover, similar findings regarding the safety and effectiveness of the flu vaccine were reported by other studies [11, 16].

Although participants were more knowledgeable about influenza and the need for a vaccine, they still had harmful misconceptions about the disease and immunisations. For example, over half of those questioned who were not vaccinated felt that they not only had alternative ways to protect themselves from the infection but that the vaccine had serious adverse effects; furthermore, over 47.4% though it was not effective anyway, with almost that many stating the flu is a minor disease which does not require immunisation. When considering the attitudes and perceptions of participants, only a little over half agreed that it was vital to get the flu vaccine every year, with even fewer reporting they would recommend its use for everyone. More participants believed that vaccination could be used to thwart dangerous health-related problems associated with the flu, with over three-quarters stating they would get one if it were proven effective in preventing seasonal flu. Unfortunately, these perceptions and attitudes resulted in unhealthy beliefs, as 40.4% of those responding stated they did not need a vaccine since they were already immune. Overall, only a little more than half (52%) of people displayed positive attitudes regarding immunisation. Furthermore, the results indicated that participants with a history of previous vaccination had a significantly higher level of knowledge and positive attitudes, which resulted in increased vaccination coverage.

In this study, the rate those ever having received influenza vaccination among participants was 43.3%. A similar finding was reported in similar studies that targeted the general Saudi

population. The prevalence rates were 44.53%, 36.7%, and 55% in studies by Alqahtani et al. (2017), Sagor and AlAteeq (2018), and Alljamili (2020), respectively [11, 14, 16]. Our study found the main sources of information that influenced participants to undergo vaccination were doctor's advice and educational campaigns. Such a finding was in accordance with Alqahtani et al. (2017), who reported that HCWs were the main source of information [11]. However, other studies ranked mass media as the first and HCWs as the third source of vaccine information for the general Saudi population [14, 17].

The results of this study confirm previous studies available in the literature. As Masadeh et al. (2014) found regarding mothers' attitudes and beliefs, those with negative perceptions of vaccinations were more likely to have not only less knowledge, but also lower immunisation rates for their children [23]. The study by Mapatano et al. (2008) was intriguing, as it indicated both support and well as contradictory evidence for study the study by Masadeh et al. [23]. Although almost all mothers had a positive attitude about the benefits of vaccination, their knowledge was severely lacking (especially regarding which conditions could vaccines protect their children from); only a little more than one-third of children received vaccinations [24]. However, these results depended upon the geographical area and whether it was in a location of low- or high-vaccination coverage: although mothers' knowledge was positively associated with immunisation in low-coverage areas, these vaccination rates also depended upon fathers' education levels in high-coverage zones [24]. Moreover, our study found that 28.8% of our participants believed that the influenza vaccine was not necessary to keep them free from flu; such a finding was consistent with those of Bukhsh et al. (2018), who reported that 35% of Pakistani parents did not think that these vaccines were required for their children to be healthy [25]. This study provided a unique perspective to this research.

Another critical population is school teachers, as they are in close contact with children (in areas where points of infection could arise) and have a significant influence on their education and, therefore, public health. In Riccò et al.'s 2017 study, over 67% of teachers stated that their motivation for vaccination was so they would not become infected; however, their knowledge regarding the specifics of how vaccines protect people, as well as of seasonal influenza, indicated a substantial knowledge gap, which had shaped their perceptions and attitudes [26]. Similar results were found in a study on workers in China. Although many respondents felt that vaccinations were effective in preventing flu, only 6.5% were aware of its timeframe; this had serious consequences for the voluntary vaccination rate, which was only a little less than 24% [27]. Many who answered the questionnaires believed mistakenly that they were strong enough to fight off the flu without the need for a vaccine. This seems to indicate that, even when there are positive attitudes about vaccinations, knowledge gaps are sufficient to overcome these perceptions and result in reduced vaccination coverage. Again, the study by Ermenlieva et al. (2019) came to similar conclusions, where a lack of knowledge was linked with low vaccination rates regardless of the perceptions and attitudes being primarily positive [28].

In the literature review of studies specifically in Saudi Arabia, researchers came to similar conclusions. In Sagor and AlAteeq's 2018 study, those with prior knowledge of the influenza vaccination had a much greater likelihood of receiving the vaccination; furthermore, being aware of the vaccinations being free contributed to higher vaccination rates [14]. Also, such findings were supported by a systematic review that showed that lack of knowledge is the chief barrier to influenza vaccination [15]. These findings were corroborated by Alqahtani et al.'s 2017 study, which confirmed that knowledge regarding vaccine immunisation is strongly linked with the way individuals decide to proceed with vaccinations: those who were previously vaccinated displayed more knowledge regarding both seasonal influenza and the flu vaccine [11]. Those who were less likely to be vaccinated had negative perceptions, mistakenly

believing that the vaccine weakens the immune system [11]. A study by Alabbad et al. (2018) reported that the most common reason to refuse the influenza vaccine among the Saudi public was their belief that the vaccine was of no benefit, that they were healthy, and that the vaccine caused serious side effects [18]. Meanwhile, a study by Sales et al. (2021) showed that Saudi citizens who believed vaccination to be safe, efficacious and given at a specific time of year, and were aware of the need to be vaccinated, were more likely to have received the flu vaccine [17]. These results also support the current study, where any adverse beliefs or attitudes were associated with reduced vaccination coverage. In Saudi Arabia, even though HCWs may have more knowledge regarding vaccinations (as well as an understanding that they are more at risk since they are in close proximity to ill individuals), negative perceptions still have a significant influence on whether they received vaccination; an obstacle to immunisation is their concern regarding its safety and the associated adverse effects [19, 20]. Meanwhile, studies reported that awareness of the risk of infection and the need for protection was the main motivator for HCWs in Saudi Arabia to be vaccinated [19, 20]. These studies also support the current research, as negative attitudes, especially when combined with knowledge gaps, resulted in a reduced likelihood of a person receiving a vaccination.

## Limitations

The study only occurred at four primary healthcare facilities within Riyadh, Saudi Arabia: Alsulimania, Alwady, Alyasmin, and Alsahafah. Therefore, it was geographically limited. This may make it more difficult to generalize the results to other populations. Furthermore, it only included those individuals (patients, family members, and visitors) who visited the facilities within the study's period of implementation. This also makes the resulting data challenging to generalize. Also, because most participants were male, employed, held college or university degrees, extra caution should be exercised in extrapolating our findings to the entire Saudi population. Unlike earlier studies, this study was conducted during a respiratory pandemic caused by Corona Virus Disease-2019 (COVID-19). This may bias the results in favor of certain directions. Moreover, the cross-sectional design of the study precludes the causality.

## Recommendations

For individuals in Saudi Arabia, receiving the seasonal influenza vaccination is an effective and safe way to limit the burden of the disease. However, as recent research shows, vaccination coverage is still suboptimal in the country, with KAPs factors being associated with vaccination rates. According to our findings and those of Ermenlieva et al. (2019), when individuals lack information and a negative attitude, they are less likely to receive vaccination; therefore, having sufficient knowledge and a positive attitude are positively correlated with being vaccinated [28]. By increasing education regarding influenza and vaccinations, people in Saudi Arabia will become more informed regarding the potential benefits of vaccination as well as the consequences of not being immunised. Along with the flu vaccine campaign, a systematic public awareness programme on seasonal influenza and influenza vaccinations are advised to be undertaken yearly. Furthermore, HCWs are advised to discuss with their patients the importance of getting a flu vaccine regularly, particularly for those at risk. This will also shape attitudes and perceptions, hopefully resulting in a more positive perspective regarding influenza vaccinations.

## Conclusion

By questioning patients, family members, and visitors attending primary healthcare centers in Riyadh city throughout the implementation period, this study found that those who lacked

knowledge regarding seasonal influenza, or the influenza vaccine were more likely not only to have negative attitudes and perceptions regarding the vaccine but also not to receive one. Roughly half of the respondents showed positive attitudes regarding immunisation, meaning that half had pessimistic viewpoints, attributing these to adverse side effects; furthermore, even those with optimistic beliefs about vaccinations still mistakenly believed they did not need them (i.e., they had other protections). This indicates a significant knowledge gap in Riyadh's citizens, necessitating educational strategies to inform them of the overwhelming benefits of immunisation. As this study concludes, respondents with more accurate, reliable knowledge enjoyed associated positive attitudes and were subsequently more likely to be vaccinated. These are encouraging findings, indicating numerous advantages for Riyadh in improving education and informing citizens about influenza and influenza vaccinations.

## Supporting information

**S1 Data.**
(XLSX)

## Author Contributions

**Data curation:** Ola Alqudah.

**Formal analysis:** Ahmad M. Al-Bashaireh.

**Investigation:** Ola Alqudah.

**Methodology:** Ola Alqudah.

**Supervision:** Ahmad M. Al-Bashaireh.

**Validation:** Ahmad M. Al-Bashaireh.

**Writing – original draft:** Norah Alhatim.

**Writing – review & editing:** Norah Alhatim, Ahmad M. Al-Bashaireh.

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
