## [Decision Letter · Decision Letter 0]

24 Jun 2021

PONE-D-21-12775

Knowledge, attitude, and practice of seasonal influenza and influenza vaccine immunization among people visiting primary healthcare centers in Riyadh, Saudi Arabia

PLOS ONE

Dear Dr. Alqudah,

Thank you for submitting your manuscript to PLOS ONE. After careful consideration, we feel that it has merit but does not fully meet PLOS ONE’s publication criteria as it currently stands. Therefore, we invite you to submit a revised version of the manuscript that addresses the points raised during the review process.

We look forward to receiving your revised manuscript.

Kind regards,

Livia Melo Villar

Academic Editor

PLOS ONE

Journal Requirements:

Additional Editor Comments (if provided):

Dear Author,

This manuscript intends to assess the knowledge, attitudes, and practices of seasonal influenza and influenza vaccine immunization among clients visiting healthcare centers in Riyadh, Saudi Arabia.

This information is very useful to recognize gaps to increase vaccine coverage in this population. I suggest the revision of the paper as reviewers comments to increase the relevance and quality of the paper,

sincerely,

Reviewers' comments:

Reviewer's Responses to Questions

**Comments to the Author**

1. Is the manuscript technically sound, and do the data support the conclusions?

Reviewer #1: Yes

Reviewer #2: Yes

Reviewer #3: No

2. Has the statistical analysis been performed appropriately and rigorously? 

Reviewer #1: Yes

Reviewer #2: Yes

Reviewer #3: Yes

3. Have the authors made all data underlying the findings in their manuscript fully available?

Reviewer #1: No

Reviewer #2: Yes

Reviewer #3: Yes

4. Is the manuscript presented in an intelligible fashion and written in standard English?

Reviewer #1: Yes

Reviewer #2: Yes

Reviewer #3: Yes

5. Review Comments to the Author

Reviewer #1: Main questions/comments:

- Is there a program also for the elderly and healthcare workers? Could you explain more about the Influenza vaccination program in Saudi Arabia, the target groups, numbers of individuals and access to vaccine (public or private?)

- Data on literature presented from lines 112 to 221 could be transferred to the discussion section, comparing to the data you generated in the article, or make it shorter.

- Were the answers to the questions about Knowledge and Practices with multiple choices or short answers? Please explain that.

- On table 3 it is reported that 356 individuals answered yes to the question “Does the vaccine have side effects?” However, on the lines below, 414 and 398 participants stated that vaccine causes Soreness and Fever, respectively. Please, check these numbers.

- Please make it clearer what was the knowledge added to the field of your study compared to Sagor and AlAteeq’s (2018) Alqahtani et al.’s (2017) and study.

English review:

Page 1 line 11: “The study investigated the relationship…” instead of “will investigate”

Line 21 “43.7% of participants had ever received” instead of “were ever received”

Page 2, line 49: the correct is “flu epidemics” not endemics

Line 55: “influenza infections across the world affects an average of five million people” sounds better than the original sentence

Line 64: find a better reference for this sentence, a scientific paper instead of Medscape

Line 70: use “its development” instead of “their development”

Page 3 line 71-72: please reformulate the sentence “The goal of the influenza vaccine is to protect against disease and achieve high vaccination rates” The vaccine itself doesn’t have the goal of achieving high vaccination rates.

Line 77: remove the word “scare”

Page 12 line 257: change “the questionnaire was clear and shows an acceptable level of internal reliability” to “the questionnaire was clear and showed an acceptable level of internal reliability”

Line 272 Please change “3. Have you ever heard of a vaccine prevent flu?” to “3. Have you ever heard that a vaccine could prevent flu?”

Line 388: change “There were 43.7% of participants (267 out of 611) were ever received flu vaccine ” to “There were 43.7% of participants (267 out of 611) that ever received flu vaccine”

Reviewer #2: The manuscript titled "Knowledge, attitude, and practice of seasonal influenza and influenza vaccine immunization among people visiting primary healthcare centers in Riyadh, Saudi Arabia" by Norah Alhatim, Ahmad M. Al-Bashaireh and Ola M. Alqudah presents a cross-sectional, descriptive and correlational study to assess the knowledge, attitudes, and practices (KAP) of seasonal influenza and influenza vaccine immunization among four Riyadh’s primary healthcare centers.

Through the data found, it is clear the need to invest in public awareness policies to increase population adherence in vaccination campaigns in order to mitigate potential new epidemics.

The lack of basic and scientific knowledge, combined with religious factors, can contribute to vaccine-refractory attitudes. This theme, when extrapolated to the Covid pandemic, shows the importance of investing in vaccine adherence campaigns with the support of local competent authorities.

Reviewer #3: The work presented here by Alhatim et al. brings a very interesting data concerning knowledge, attitude, and practice of seasonal influenza and influenza vaccine immunization, in Saudi Arabia. Although, this study showed concise results I feel that the manuscript needs to be improved and a wider revision on the theme need to be done, so the article can benefit greatly from a deeper immersion on currently published works. In this sense, some of my suggestions are listed below.

Major points:

In my opinion the manuscript needs to be actualized, benefiting greatly with the inclusion of several studies listed in this revision (discussion section below).

- Introduction

In my opinion some paragraphs can be removed and other can be presented in a simple and lean way. Paragraphs from line 141 to 183 should be removed and the ones from lines 184 to 228 can be moved to the discussion section in a more concise form.

- Discussion

As mentioned before it needs to incorporate related studies that will give the reader a better understand of your results and the published literature in similar subjects.

Discussion and introduction sections will benefit greatly from the following articles:

Aljamili AA. Knowledge and practice toward seasonal influenza vaccine and its barriers at the community level in Riyadh, Saudi Arabia. J Family Med Prim Care. 2020 Mar 26;9(3):1331-1339. doi: 10.4103/jfmpc.jfmpc_1011_19.

Alabbad AA, Alsaad AK, Al Shaalan MA, Alola S, Albanyan EA. Prevalence of influenza vaccine hesitancy at a tertiary care hospital in Riyadh, Saudi Arabia. J Infect Public Health. 2018 Jul-Aug;11(4):491-499. doi: 10.1016/j.jiph.2017.09.002.

Alotaibi FY, Alhetheel AF, Alluhaymid YM, Alshibani MG, Almuhaydili AO, Alhuqayl TA, Alfayez FM, Almasabi AA. Influenza vaccine coverage, awareness, and beliefs regarding seasonal influenza vaccination among people aged 65 years and older in Central Saudi Arabia. Saudi Med J. 2019 Oct;40(10):1013-1018. doi: 10.15537/smj.2019.11.24587.

Sales IA, Syed W, Almutairi MF, Al Ruthia Y. Public Knowledge, Attitudes, and Practices toward Seasonal Influenza Vaccine in Saudi Arabia: A Cross-Sectional Study. Int J Environ Res Public Health. 2021 Jan 8;18(2):479. doi: 10.3390/ijerph18020479.

Zaraket H, Melhem N, Malik M, Khan WM, Dbaibo G, Abubakar A. Review of seasonal influenza vaccination in the Eastern Mediterranean Region: Policies, use and barriers. J Infect Public Health. 2020 Mar;13(3):377-384. doi: 10.1016/j.jiph.2020.02.029.

Balkhy HH, Abolfotouh MA, Al-Hathlool RH, Al-Jumah MA. Awareness, attitudes, and practices related to the swine influenza pandemic among the Saudi public. BMC Infect Dis. 2010 Feb 28;10:42. doi: 10.1186/1471-2334-10-42.

Alshammari TM, Yusuff KB, Aziz MM, Subaie GM. Healthcare professionals' knowledge, attitude and acceptance of influenza vaccination in Saudi Arabia: a multicenter cross-sectional study. BMC Health Serv Res. 2019 Apr 15;19(1):229. doi: 10.1186/s12913-019-4054-9.

Awadalla NJ, Al-Musa HM, Al-Musa KM, Asiri AM, Albariqi AA, Majrashi HM, Alasim AA, Almuslah AS, Alshehri TK, AlFlan MA, Mahfouz AA. Seasonal influenza vaccination among primary health care workers in Southwestern Saudi Arabia. Hum Vaccin Immunother. 2020;16(2):321-326. doi: 10.1080/21645515.2019.1666500.

Alolayan A, Almotairi B, Alshammari S, Alhearri M, Alsuhaibani M. Seasonal Influenza Vaccination among Saudi Children: Parental Barriers and Willingness to Vaccinate Their Children. Int J Environ Res Public Health. 2019 Oct 31;16(21):4226. doi: 10.3390/ijerph16214226.

Balaban V, Stauffer WM, Hammad A, Afgarshe M, Abd-Alla M, Ahmed Q, Memish ZA, Saba J, Harton E, Palumbo G, Marano N. Protective practices and respiratory illness among US travelers to the 2009 Hajj. J Travel Med. 2012 May-Jun;19(3):163-8. doi: 10.1111/j.1708-8305.2012.00602.x.

Alqahtani AS, Rashid H, Heywood AE. Vaccinations against respiratory tract infections at Hajj. Clin Microbiol Infect. 2015 Feb;21(2):115-27. doi: 10.1016/j.cmi.2014.11.026.

Minor points:

Abstract

Page 1 line 2 and 3: rephrase the following sentence: “Influenza infection is a continuing threat to the Saudi population, resulting in high mortality rates and disease prevalence; it also puts significant financial strain on the country.”

Introduction

This section needs to be shortened and actualized (please see articles suggested at the discussion section).

Page 3 line 57: remove: “However, it affects all age groups and is usually self-limited [4]”

Materials and Methods

Page 11 - Line 236-239: Informe the period (years) of data collection.

Page 12/13 - Line 262 -293: Variables and Domains should be presented in a box instead of being listed in the text. This way will be easier for the reader to have access to the information.

Results

Page 15 line 331: “When asked about the flu”

Page 15 line 337: Table 1. Socio-demographic characteristics (N = 611).

Page 18 line 362-370: Tables 4 to 7 should be merged in a unique table with all information.

Page 19: Table 9 should be excluded the information is already on the text. This Table brings no additional information to the reader, this suggestion is also applicable for Tables 11, 12 and 14 on page 20, 21 and 22, respectively.

Discussion

See Major points above.

Recommendations

It’s very important to include here proposals of how increased education regarding influenza and vaccinations can be achieved, which activities need to be better addressed and which should be strengthen.

Conclusion

Page 26 - Lines 520 to 523: remove

6. PLOS authors have the option to publish the peer review history of their article (what does this mean?). If published, this will include your full peer review and any attached files.

Reviewer #1: No

Reviewer #2: **Yes: **Yasmine Rangel Vieira

Reviewer #3: No

---

## [Author Response · Author response to Decision Letter 0]

19 Oct 2021

All proposed modifications have been taken into account and modified. Thank you for enriching our research.

---

## [Decision Letter · Decision Letter 1]

25 Nov 2021

PONE-D-21-12775R1Knowledge, attitude, and practice of seasonal influenza and influenza vaccine immunization among people visiting primary healthcare centers in Riyadh, Saudi ArabiaPLOS ONE

Dear Dr. Alqudah,

Thank you for submitting your manuscript to PLOS ONE. After careful consideration, we feel that it has merit but does not fully meet PLOS ONE’s publication criteria as it currently stands. Therefore, we invite you to submit a revised version of the manuscript that addresses the points raised during the review process. Please address reviewer 1's comments. Please also address the following requests:In your methods section, please provide additional information about the participant recruitment method and the demographic details of your participants. In particular, please include a statement as to whether your sample can be considered representative of a larger population. In addition, please further discuss as a limitation in the Results or Conclusion section the non-representativeness of your sample (in particular the relatively high rate of respondents who attended college).*PLOS ONE* does not copy edit accepted manuscripts. Please proofread for typos and grammar, including in the abstract.

We look forward to receiving your revised manuscript.

Kind regards,

Yann Benetreau, PhD

Senior Editor (Staff editor), *PLOS ONE*

Journal Requirements:

Reviewers' comments:

Reviewer's Responses to Questions

**Comments to the Author**

1. If the authors have adequately addressed your comments raised in a previous round of review and you feel that this manuscript is now acceptable for publication, you may indicate that here to bypass the “Comments to the Author” section, enter your conflict of interest statement in the “Confidential to Editor” section, and submit your "Accept" recommendation.

Reviewer #1: All comments have been addressed

Reviewer #2: All comments have been addressed

2. Is the manuscript technically sound, and do the data support the conclusions?

Reviewer #1: Yes

Reviewer #2: Yes

3. Has the statistical analysis been performed appropriately and rigorously? 

Reviewer #1: Yes

Reviewer #2: Yes

4. Have the authors made all data underlying the findings in their manuscript fully available?

Reviewer #1: Yes

Reviewer #2: Yes

5. Is the manuscript presented in an intelligible fashion and written in standard English?

Reviewer #1: Yes

Reviewer #2: Yes

6. Review Comments to the Author

Reviewer #1: I think the review answered all the questions and points I had raised and the new version has been improved substantially. Just another question that arose when I read the reviewed manuscript. The data collection for this study was from January to July 2020. It would be important to include one or two sentences discussing the impact of Covid-19 pandemics on the results obtained in this study, compared to previous years, without the threat of a respiratory pandemic.

Reviewer #2: (No Response)

7. PLOS authors have the option to publish the peer review history of their article (what does this mean?). If published, this will include your full peer review and any attached files.

Reviewer #1: No

Reviewer #2: No

---

## [Editor Report · Decision Letter 2]

22 Mar 2022

Knowledge, attitude, and practice of seasonal influenza and influenza vaccine immunization among people visiting primary healthcare centers in Riyadh, Saudi Arabia

PONE-D-21-12775R2

Dear Dr. Alqudah,

We’re pleased to inform you that your manuscript has been judged scientifically suitable for publication and will be formally accepted for publication once it meets all outstanding technical requirements.

Sincerely yours,

Yann Benetreau, PhD

Division Editor, PLOS ONE
---

## [Editor Report · Acceptance letter]

24 Mar 2022

PONE-D-21-12775R2 

Knowledge, attitude, and practice of seasonal influenza and influenza vaccine immunization among people visiting primary healthcare centers in Riyadh, Saudi Arabia 

Dear Dr. Alqudah:

I'm pleased to inform you that your manuscript has been deemed suitable for publication in PLOS ONE. Congratulations! Your manuscript is now with our production department. 

Kind regards, 

on behalf of

Dr. Yann Benetreau 

Staff Editor

PLOS ONE